# Binding-induced functional-domain motions in the Argonaute characterized by adaptive advanced sampling

**Danial Pourjafar-Dehkordi**, **Martin Zacharias** *

Center of Functional Protein Assemblies, Technische Universität München, Garching, Germany

* zacharias@tum.de

**Data Availability Statement:** All relevant data are within the manuscript and its Supporting Information files.

## Abstract

Argonaute proteins in combination with short microRNA (miRNAs) can target mRNA molecules for translation inhibition or degradation and play a key role in many regulatory processes. The miRNAs act as guide RNAs that associate with Argonaute and the complementary mRNA target region. The complex formation results in activation of Argonaute and specific cleavage of the target mRNA. Both the binding and activation processes involve essential domain rearrangements of functional importance. For the *Thermus Thermophilus* Argonaute (TtAgo) system guide-bound (binary) and guide/target-bound (ternary) complexes are known but how the binding of guide and target mediate domain movements is still not understood. We have studied the Argonaute domain motion in apo and guide/target bound states using Molecular Dynamics simulations and a Hamiltonian replica exchange (H-REMD) method that employs a specific biasing potential to accelerate domain motions. The H-REMD technique indicates sampling of a much broader distribution of domain arrangements both in the apo as well as binary and ternary complexes compared to regular MD simulations. In the apo state domain arrangements corresponding to more compact (closed) states are mainly sampled which undergo an opening upon guide and guide/target binding. Whereas only limited overlap in domain geometry between apo and bound states was found, a larger similarity in the domain distribution is observed for the simulations of binary and ternary complexes. Comparative simulations on ternary complexes with 15 or 16 base pairs (bp) formed between guide and target strands (instead of 14) resulted in dissociation of the 3'-guide strand from the PAZ domain and domain rearrangement. This agrees with the experimental observation that guide-target pairing beyond 14 bps is required for activation and gives a mechanistic explanation for the experimentally observed activation process.

## Author summary

Post-transcriptional gene silencing is an important process to regulate protein synthesis in eukaryotes and prokaryotes. The Argonaute proteins as part of the RNA-induced-silencing-complex (RISC) form a central element of the process by silencing of a target

**Funding:** MZ received funding by German Research Foundation DFG, Sonderforschungsbereich 1035, Projektnummer 201302640, project B02. www.dfg.de. Computational resources were provided by the Leibniz Supercomputing Center (LRZ) within grant pr27za. The funders had no role in study design, data collection and analysis, decision to publish, or preparation of the manuscript.

**Competing interests:** The authors have declared that no competing interests exist.

messenger RNA (mRNA) via degradation or repression of translation. The Argonaute protein binds initially a short RNA that acts as a guide to promote binding of a complementary target mRNA. The complex formation can lead to activation of Argonaute and specific cleavage of the target mRNA. The whole process involves domain rearrangements that are not fully understood. We applied an advanced Molecular Dynamics sampling technique to specifically accelerate domain motions of the *Thermus Thermophilus* Argonaute (TtAgo) system in apo, guide bound and guide/target bound states. The simulations indicate only limited overlap of domain arrangements in apo and bound states and identified domain opening motions necessary for guide and target binding. The study also offers an explanation why a minimum of 15 or 16 base pairs between guide and target strands are necessary for Argonaute activation.

## Introduction

The Argonautes constitute a family of proteins that are involved in both transcriptional and post-transcriptional gene regulatory mechanisms [1–3]. They are present in all forms of life, sharing a well conserved tertiary structure, despite their limited sequence similarity [4–6]. Argonautes are key components of the RNA-induced Silencing Complex (RISC), which drives silencing of a target messenger RNA (mRNA) via degradation or repression. In eukaryotes, the RISC complex is guided by a single-stranded RNA that makes base-pairing with the target. The guide RNA can originate from single microRNA (miRNA) or short interfering RNA (siRNA) pathways [1]. The regulation of gene expression via miRNAs is different from that of the siRNA, as the latter binds to a fully complementary target mRNA and therefore, prompts specific gene silencing, while the former can regulate multiple mRNAs. Prokaryotic Argonautes, on the other hand, participate in gene regulation by binding to single-stranded DNAs that guide them towards target DNAs or RNAs that are either cleaved or repressed [7–10]. Moreover, bacterial Argonaute protects its host against invasive genomic elements through directly targeting foreign DNA molecules [9,11].

The overall structure of most Argonautes features four characteristic domains. The N-terminal domain (N) functions as a wedge to unwind guide-target duplexes [3,5,12–14]. The P-element-induced wimpy testis (PIWI)-Argonaute-Zwille (PAZ) domain hosts the binding pocket for the 3' region of the guide strand [14,15]. The phosphorylated 5' end of the guide is stabilized at the interface between the middle (MID) and PIWI domains via interaction with a $Mg^{2+}$ cation [5,15–17]. The four domains are distributed in two lobes with PAZ and N in the one lobe and MID and PIWI in the other lobe. The lobes are connected by two linker domains, L1 and L2 [5,14,18].

In a series of structural and biochemical studies Patel and coworkers have determined the structure of the *Thermus Thermophilus* Argonaute (TtAgo) in several guide-bound (binary) and guide/target-bound (ternary) complexes–varying in duplex length and level of duplex complementarity [11,12,15,19]. These structural snapshots portrayed the protein in various stages of the silencing process. The structures of Argonaute proteins from other organisms have also been studied [7,20], however, not as extensively as TtAgo. The structural studies reveal that both 5' and 3' ends of the 21-mer guide DNA strand are anchored in corresponding binding pockets in the MID and PAZ domains respectively. The 3' end binding to the PAZ domain is characteristic of the *inactive* domain arrangement, which is stabilized in the presence of complementary target strands with a length below 15 nucleotides. Upon target binding followed by base pairing beyond position 16 of the guide strand, a switch to the cleavage-

compatible conformation–known as the *active* state occurs, in which the 3' end is dissociated from PAZ. The active state is distinguished by a tetrad of DEDD amino-acids residues in the PIWI domain that attacks the cleavage site between positions 10 and 11 of the target strand [17]. Moreover, with the detachment of the 3' end of the guide, a pivot-like movement of the PAZ domain towards the MID domain is observed [12,15,17,19]. Single-molecule fluorescence resonance energy transfer (smFRET) studies of the active-complex formation demonstrated the dynamic rearrangements of PAZ and the 3' end of the nucleic acid bound to it [21,22]. Based on the structural, biochemical and biophysical studies a mechanistic stepwise scheme has evolved with an initial binding of the guide strand to TtAgo, anchoring of the 5' and 3' ends followed by binding of a complementary RNA or DNA strand that–if long enough–leads to dissociation of the guide's 3' end from the PAZ domain, enzyme activation and finally, cleavage of the target. The dissociation of the cleaved target and rebinding of the 3' end to PAZ leads again to the initial inactive guide-bound state. The critical role of the PAZ domain anchoring of the 3' end of the guide strand to inactivate the enzyme is further supported by experiments employing short guide DNA or RNA molecules [12,15]. Even guide RNAs as short as 9 nucleotides which do not allow the anchoring of the 3' end at the PAZ domain result in efficient cleavage of target RNA/DNA strands [15]. Furthermore, kinetic observations of guide and target RNAs binding to human Argonaute-2 (hAgo2), a homolog of TtAgo, revealed that the 3' end's release from PAZ is the rate-limiting step during the activation process [23]. Despite previous structural and biochemical studies, a detailed atomistic view of the extension of the base pairing between guide and target strands leading to the dissociation of the 3' end and subsequent domain rearrangements and enzyme activation has remained elusive.

TtAgo adopts at least two conformational states (i.e., inactive and active) and the transition between these two states involves domain rearrangements, and it is likely that there are additional intermediate states. The available crystal structures of TtAgo in complex with different guide and/or target strands give an excellent overview on stable domain arrangements, however, do not provide insight into the global states accessible for apo TtAgo or binary and ternary complexes. In principle, Molecular Dynamics (MD) simulations can provide a dynamic high-resolution view on possible structural arrangements of TtAgo. However, crossing the energy barriers separating metastable conformational states, especially when it involves an interplay between a multidomain protein and nucleic acids, may require simulations times well beyond currently accessible MD timescales. Most previous MD simulation studies of Ago proteins were focused on the nucleic acid recognition and binding process and the corresponding structural changes in hAgo2 [24] and TtAgo [25]. In studies, the combination of MD with bias-exchange metadynamics [26] and protein-DNA docking methods revealed an induced-fit mechanism of the guide-strand loading in TtAgo [27]. In hAgo2, however, a two-step mechanism RNA loading was suggested, based on a study that employed Markov State Models and protein-RNA docking [28].

In the present study we employ a replica-exchanged based enhanced-sampling MD method to specifically accelerate domain motions in the TtAgo system in different DNA-bound states and in the absence of any substrates (apo state). In our approach a biasing potential is constructed by a mixture of Gaussians–similar to a metadynamics simulation–along center-of-mass distance variables. The form and level of biasing in each replica is rapidly adjusted during an equilibration phase and integrated in the force field of the Hamiltonian replica-exchange molecular dynamics (H-REMD) simulation. It allows exploration of regions in the global conformational landscape separated by energy barriers that are not sampled in regular MD simulations. In contrast to a metadynamics approach, no multidimensional (slowly adapting) biasing is required but biasing in separate distance variables is used during H-REMD. In addition to characterizing the domain mobility, in the apo state, in the binary as well as in the

ternary complexes, we also apply the methodology to reveal the structural transitions that steer the protein from an inactive to an active conformation. In agreement with the experimental observations, we detect the dissociation of the 3' end of the guide strand from the PAZ domain when extending the target/guide duplex beyond a critical length of 14 base pairs, due to sterical strain. The dissociated 3'-end ultimately settles in the cleft between N and PIWI domains but remains conformationally highly mobile. The associated conformational substates and inter-mediates are also investigated.

## Materials and methods

### Thermus Thermophilus Argonaute starting structure

The protein start structure was taken from PDB-entry 4n41 corresponding to the ternary TtAgo complex with 21-mer guide DNA and 15 nucleotides of target DNA [17]. The DNA duplex in the crystal structure is resolved in positions 1–14 and 20, 21 in the guide strand and 1–14 in the target strands. All structurally resolved nucleotides were kept and restrained during the equilibration phases of the simulations. The missing bases–position 15–19 of the guide were added using in part other TtAgo structures and using the MODELLER software [29]. In case of simulations with target strand extension to yield 15 or 16 base pairs with the guide strand were added to the DNA in the PDB 4n41 structure in B-form and Watson-Crick base pair geometry. To further stabilize the base-pairing geometry distance restraints to keep WC base pairing were imposed during an equilibration phase of the simulations. The ff14SB force field and TIP3P water model were used to model proteins and the explicit solvent molecules [30,31]. The parameters for the phosphorylated 5' end of the guide strand were generated using the generalized Amber force field (gaff) [32]. The OL15 force field refinements were employed for nucleic acids [33]. The solvated box was then energy minimized (2500 steps), followed by 25 ps of heating and 50 ps of density equilibration in an NPT ensemble with the pressure kept at 1 bar and temperature adjusted to 300 K using a 2 fs time step. During these phases, the protein's heavy atoms, the nucleotide and the magnesium ions were restrained at their initial positions using a harmonic potential with a decreasing force constant, starting at 5.0 kcal·mol·Å$^{-2}$ and ending with 1.0 kcal·mol·Å$^{-2}$. The solvated box was equilibrated further in a restraint-free NPT ensemble at 300 K and 1 bar for 1 ns. The temperature for the unrestrained data-gathering production MD simulations was 315 K. The GPU-accelerated pmemd. cuda version of the Amber 18 software package[34] was used implementing the hydrogen mass repartitioning feature which allows a simulation time step of 4 fs [35]. Long range interactions were included using the particle mesh Ewald (PME) method combined with periodic boundary conditions and a 9 Å real space cut-off. Trajectory analysis was performed with the cpptraj modul of Amber18. Figures were generated using the PyMol software package [36].

**Replica-exchange simulation protocol.** The simulation setup employed version 18 of the Amber software package and a python library. The Amber pmemd.cuda code was modified to accommodate a Gaussian-shaped bias potential between center-of-mass (COM) distances in the GPU version. The replica-exchange simulations were initialized using a batch file that contained all the input parameters, including the number of replicas, time steps and exchange attempts per window, the residue IDs involved in CVs and the overall number of intervals. The term "interval" here refers to a cycle of i) replica-exchange sampling, ii) analysis of the CVs and iii) update of the biasing forces. The H-REMD simulations started with eight structurally identical replicas. The first 1 ns of simulations ran without any added bias potentials, with the aim to have an initial approximation of the distribution along the CVs in the initial phase. The trajectory files were updated every 8 ps, and the exchanges were attempted at the same time intervals. A standard Metropolis criterion was used to allow or reject the exchange

attempts. The trajectory frames of the previous 10 ns from all eight replicas were read by the python library in 1 ns intervals to calculate the CV values. The calculation of the COM distances that define the CVs and other trajectory analysis were performed using Pytraj python library [37]. Next, the library fitted the data to a Gaussian mixture model (GMM) using the sci-kit-learn machine learning package [38]. The GMM assumes that the data points are collected from a mixture of $K$ Gaussian distributions–called components–with unknown means, variances, and mixture component weights. Initial test runs indicated that using three components along each CV can efficiently reconstruct the shape of the distribution i.e., $K = 3$. For a univariate GMM with $K$ components, the $k$-th component has a mean of $\mu_k$ and variance of $\sigma_k$. The mixture component weights are defined as $\phi_k$, with the condition that $\sum_{i=1}^{k} \phi_i = 1$, i.e., the total probability distribution sums up to one. Our implementation of the GMM fits the distribution of the CVs to a weighted sum of a three-component Gaussian density, given by the equation:

$$p(x) = \sum_{i=1}^{K} \phi_i \mathcal{N}(\mathrm{x}|\mu_i, \sigma_i),$$

$$\mathcal{N}(\mathrm{x}|\mu_i, \sigma_i) = \frac{1}{\sigma_i \sqrt{2\pi}} exp\left(-\frac{(x - \mu_i)^2}{2\sigma_i^2}\right)$$

The output of the GMM–the components means, variances, and weights determine the next biasing potential during the next simulation interval. The bias potential was incremented by 4 kcal·mol$^{-1}$ in the replicas 2–8 while the reference replica ran without any added potential.

$$\sum_{r=1}^{8} B_r(\mathrm{CV}) = (r - 1) * 4\ kcal/mol * p(x)$$

In this way, each simulation is biased by an external potential $B_r(\mathrm{CV})$ that is built iteratively based on the sampling of the previous 10 ns in all replicas in such a way that it destabilizes the current sampling region. This cycle goes on until the maximum number of windows is reached (illustrated in **Fig 1**).

**Cluster analysis.** Trajectories were processed and analyzed to find similar conformations using the CPPTRAJ tool and the DBSCAN clustering algorithm [37]. We used RMSD of the protein's heavy atoms as the distance metric and used every fifth frame of the trajectories to reduce memory consumption. A minimum of 4 conformations with the distance cutoff of 1.25 Å were required to form a cluster. The initial "sieve" value was set to 10 random frames to form the initial clusters. The sieved frames were then added to the clusters as an additional step.

# Results

## Structural flexibility of Argonaute during continuous MD simulations

TtAgo is one of the best studied Argonaute proteins and, like other Argonautes, consists of PAZ, N, MID and PIWI domains that undergo domain rearrangements during the enzyme functional cycle (**Fig 2**). A goal of our study is to characterize the accessible domain geometries in the apo TtAgo, the binary (TtAgo bound to 21-mer guide DNA) and the ternary (TtAgo bound to 21-mer guide and 14-mer target DNA) complex. To examine the global domain dynamics, we first examined Argonaute's structural flexibility using extensive regular unbiased MD simulations. Since the crystal structure of TtAgo in apo form is not available, the start structure was generated by removing the DNA from the *inactive* structure of TtAgo bound to 21-mer guide and 15-mer target DNA strands (PDB: 4n41, **Fig 2**) [17]. Within ~90 ns of MD simulation time, both PAZ and N domains moved towards MID and PIWI, thereby closing

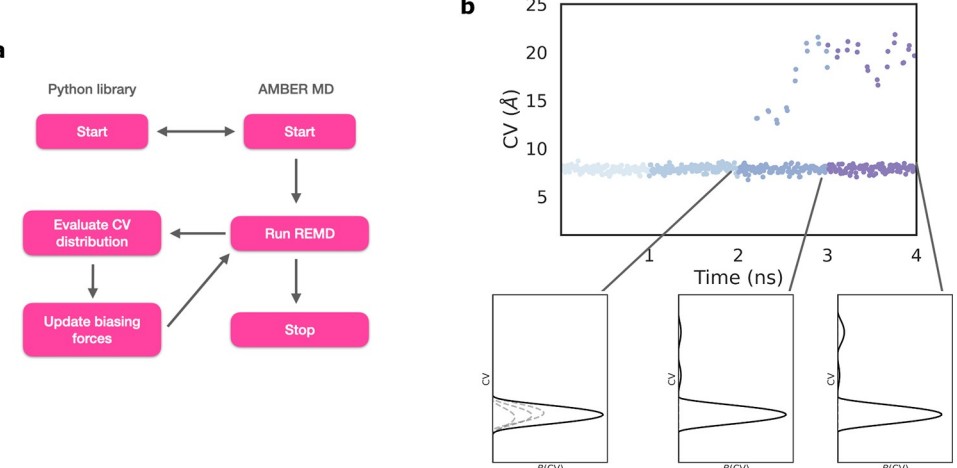

**Fig 1. Adaptive H-REMD simulation algorithm.** (**a**) The python library and the Amber MD package are initialized using a batch file that contains the input parameters. There is no biasing force inserted to the system in the first interval. At the end of each interval the trajectories are passed to the library and CVs are calculated. Based on their values in the previous runs the biasing force is updated. The H-REMD simulation then continues with the updated biasing potential introduced in the replicas. The cycle goes on until the simulation is stopped. (**b**) Illustration of the biasing potential acting on an exemplary CV vs. time. The value of CV is coloured based on the simulation interval. At the beginning the biasing along the CV is localized to the vicinity of the initially sampled regime eventually with sampling of new regions the biasing potential extends to other regions along the CV. The overall bias potential (solid black line) is the sum of three Gaussians (dashed grey lines). Note, that in the reference replica (replica 1) there is no biasing and the biasing increases incrementally with increasing replica number.

the substrate binding channel–also known as the central cleft. This arrangement is considered as a *closed* conformation and is in accordance with Argonaute's "rubber band" model. In this model, the substrate loading to the Ago protein is supported by the Hsp70/Hsp90 chaperone machinery that uses ATP to convert Argonaute from a closed to a more open structure that can accommodate the bulky strands [39]. Such opening induces structural strain in the protein, analogous to a stretched rubber band. Release of this tension in this model drives the strand separation without consuming any ATP. During unwinding, the guide strand that has the 5' end stably anchored in the pocket between MID and PIWI domains will remain in the protein, whereas the other unanchored strand–known as the passenger strand–will be discarded [39,40]. The protein and the guide strand form a functional silencing complex. Interestingly, the motion of PAZ is both translational and rotational–it translates relative to the other domains (or rotates as a whole body relative to the center of TtAgo) and also rotates around itself, while the N domain rotates only around the protein's axis (**Fig 2**).

Extending the MD simulation up to 2 μs yields a slight further downward shift of the PAZ domain (**S1 Fig**). This pivotal movement is reminiscent of the ones observed in the active TtAgo crystal structures, in which the PAZ has moved downwards and the 3' end of the guide is detached from it (PDB: 3hjf & 4nca) [12,17]. It indicates that a motion of PAZ towards the active-like geometry even in the absence of guide and target DNAs is possible. We also performed unrestrained MD simulation of the binary (with bound 21-mer guide DNA) and ternary (with bound 21-mer guide and 14-mer target DNA) complexes starting from the inactive state (PDB: 4n41, after 1 ns equilibration). Contrary to the apo form, in the binary and ternary complexes the initial domain arrangements were largely preserved during the 2 μs simulations, indicated by stable root-mean-square-deviation (RMSD) values (**S1 Fig**). In particular, the PAZ domain remained in an inactive arrangement bound

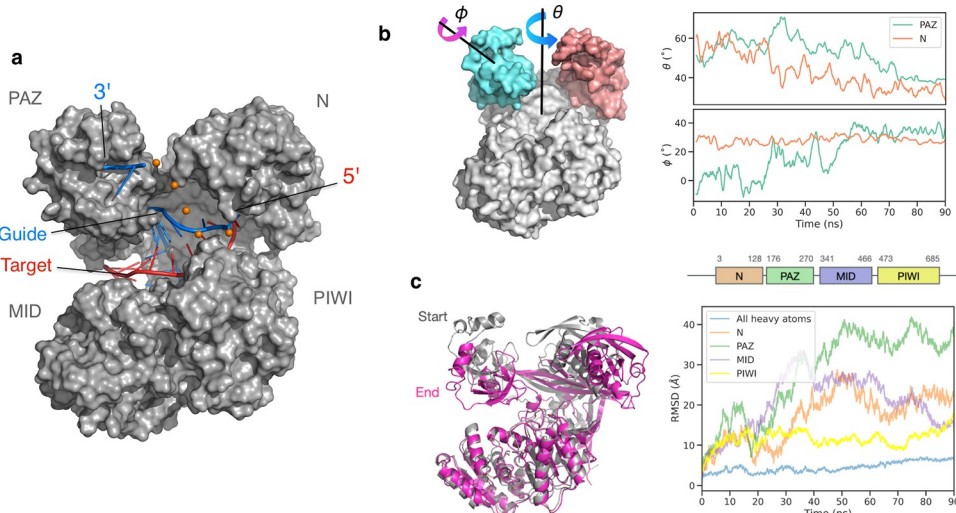

**Fig 2. Domain mobility of apo TtAgo during regular MD simulations. (a)** Crystal structure of prokaryotic Argonaute from Thermus Thermophilus bacterium (TtAgo, PDB: 4n41) bound to 21-mer guide (blue) and 15-mer target (red) DNA strands in the inactive state. The orange spheres show the presumed location of the missing bases. **(b)** *In silico* removal of the bound DNAs causes a rearrangement of the domains during unrestrained MD simulation. The angles Θ and Φ describe each domain's rotation around the protein and around itself, respectively. **(c)** The domains' RMSDs with respect to the initial structure. The last frame of the 90 ns-long free simulation served as the start structure in the following simulations of TtAgo in apo form.

to the 3' end of the guide strand. The overall stability observed in these two variants is also attributed to the hydrogen-bond network formed between the guide strand and the protein's backbone atoms [25].

## Replica-exchange simulation of TtAgo in apo, binary and ternary complexes

During regular MD simulations only limited domain motions of TtAgo were observed. Even in the apo form, after it rapidly reached a closed conformation, the protein remained stuck in that domain arrangement. It might be an artefact due to a limited sampling of the relevant conformational states on the time scale of the MD simulations. To sample putative domain arrangements more exhaustively, we employed a H-REMD technique coupled with adaptive biasing potentials along pre-selected global collective variables (CVs).

The added biasing potentials act separately on each CV, and therefore it is advantageous that the coupling between the domain movements they promote is limited. Otherwise, in the higher replicas it could lead to sampling of conformations that are of low probability or relevance for the reference replica, which runs with the original force field. To this end, we selected center-of-mass (COM) distances between domains as global variables in a hierarchal manner (S2 Fig). In TtAgo, the PIWI and MID domains are in close contact but do not directly interact with the PAZ and N domains. Hence, in the first CV we considered PIWI and MID as one unit, and PAZ and N as another unit, and the CV was defined as the distance between the COM of each of these two units ($CV_1$). The next two variables were defined as the COM distance between the N and PAZ domains ($CV_2$) and the COM distance between MID and PIWI domains ($CV_3$). For the biasing we employed one-dimensional potentials in the replicas that acted independently along the CVs (see Materials and Methods for details). During an

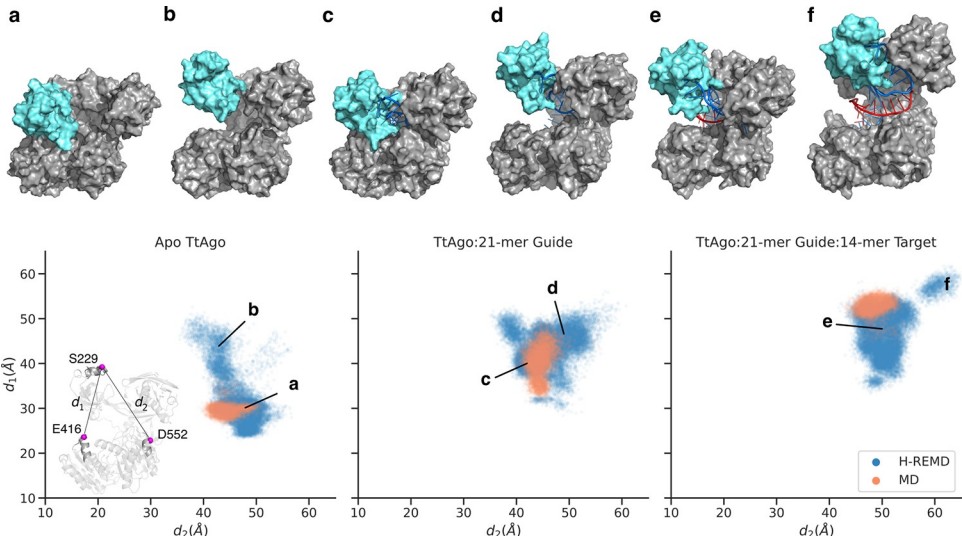

**Fig 3. Domain mobility during H-REMD with domain-domain center-of-mass distance biasing potentials.** Comparison of the PAZ domain arrangement relative to the MID and PIWI domains in 2μs unrestrained MD (orange) vs. H-REMD (blue) simulations of the apo TtAgo (left), binary (middle) and ternary (right) complexes. The placements of PAZ are indicated by the α-carbon distances $d_1$ and $d_2$ (see inset) relative to MID and PIWI domains, respectively. Snapshots in (**a**-**f**) show the corresponding arrangements marked in the plots. The PAZ domain is shown in cyan color, the guide and target strands in blue and red, respectively.

equilibration phase of the H-REMD simulations the bias potentials were adjusted to accelerate the sampling along the collective variables and the exchanges among replicas.

The H-REMD simulation of the apo TtAgo protein started from the last frame of the 90 ns unrestrained MD simulation (see above), in which the protein adopted a closed conformation. The binary (with 21-mer guide DNA) and ternary (with 21-mer guide and 14-mer target DNA) variants started from the same structure as in the unrestrained MD simulations. Evaluation of the sampled states and the trajectory analysis were done solely based on the sampling in the unbiased reference replica. A comparison of the conformational landscape of the PAZ domain in the three variants is illustrated in **Fig 3**. The plot reflects PAZ's movements relative to MID and PIWI domains, which–based on physical intuition–are measured by two α-carbon (Cα) distances, $d_1$ and $d_2$. The distance between S229 and E416 ($d_1$) represents PAZ's motion relative to MID, while the distance between S229 and E552 ($d_2$) represents PAZ's motion relative to PIWI. In all the three variants, a much broader range of the PAZ domain motion was sampled during the H-REMD simulation with an adaptive biasing potential compared to the unrestrained MD simulation. Analysis of the PAZ domain's motion at different time intervals of the H-REMD (in the reference replica) showed that the system explored all accessible domain arrangements within 160–200 ns of simulation time (**S3 Fig**). Therefore, all the replica-exchange simulations presented here were extended up to 240 ns (per replica) to assure sufficient sampling. Moreover, after this time the shape of the bias potential converged for the most fluctuating CV ($CV_2$, biasing the PAZ-N domains COM distance) (**S4 Fig**, the RMSD vs time and RMSF are illustrated in **S5** and **S6 Figs**, respectively). The approximately constant biasing potential in each replica allows then for an equilibrium sampling in each replica run including the reference replica.

Snapshots obtained from the replica-exchange simulations indicated that the apo TtAgo, in addition to the closed state, adopts conformations that are distinguished by large PAZ-MID

distances ($d_1 \approx 50$ Å). Interestingly, in the open states, the PAZ domain's position relative to MID and PIWI is similar to the low-energy states of the guide-bound TtAgo complex. The similarity in PAZ arrangements between the apo protein and the binary complex reflects the capability of the TtAgo protein to rearrange the domains and widen the central cleft in order to accommodate the bulky DNA strands. Such conformations in the apo protein, however, are energetically not favorable, due to the absence of the guide's hydrogen-bond network.

Cluster analysis of the trajectories revealed that in all variants, contrary to the upper lobe (N & PAZ), the lower lobe (MID & PIWI) has very limited flexibility (see **S7 Fig**). In guide- and guide/target-bound TtAgo, both the 3'-end and the phosphorylated 5'-end of the guide remained stably anchored in their binding pockets. Importantly, the coordination of the magnesium ion with the first phosphates of the 5'-end and the side chain of V685 of the PIWI domain was also unaltered. Previous studies have emphasized the importance of the phosphorylation of the 5'-end for the cleavage activity [7,15]. Additionally, the seed region of the guide strand (position 2–8) exhibited the lowest flexibility and remained constantly exposed to solvent (**Fig 4**). This arrangement reduces the energy barrier involved in base pairing with target strands [41]. In contrast the other single stranded segments both in the guide bound but also in the ternary complex are much more flexible and can adopt a great variety of conformations (**Fig 4**). It agrees with the experimental observation that little or no distinct electron density can be defined for these segments in crystal structures[19]. The H-REMD simulations indicate significant domain mobility especially of the PAZ domain in all studied TtAgo systems not sampled in regular MD simulations despite a longer time scale covered in the unrestrained simulations.

**Extending the target length to position 16 triggers guide's 3'-end release from PAZ.** Next, we performed H-REMD simulations of TtAgo bound to 5'-phosphorylated 21-mer guide DNA and fully complementary 15- and 16-mer target DNAs. To create the start structures, the 15th and 16th bases of the target strand were added to the initial ternary complex (PDB: 4n41), while maintaining the Watson-Crick hydrogen bonds with the opposing base on the guide strand.

Interestingly, the extension of the bound DNA duplex resulted in a wider N-PIWI gap or increased mean domain distance. The gap was recorded as the distance between the α-carbon atoms of M82 from N and D552 from PIWI and increased from ~26 Å in the 14-mer target

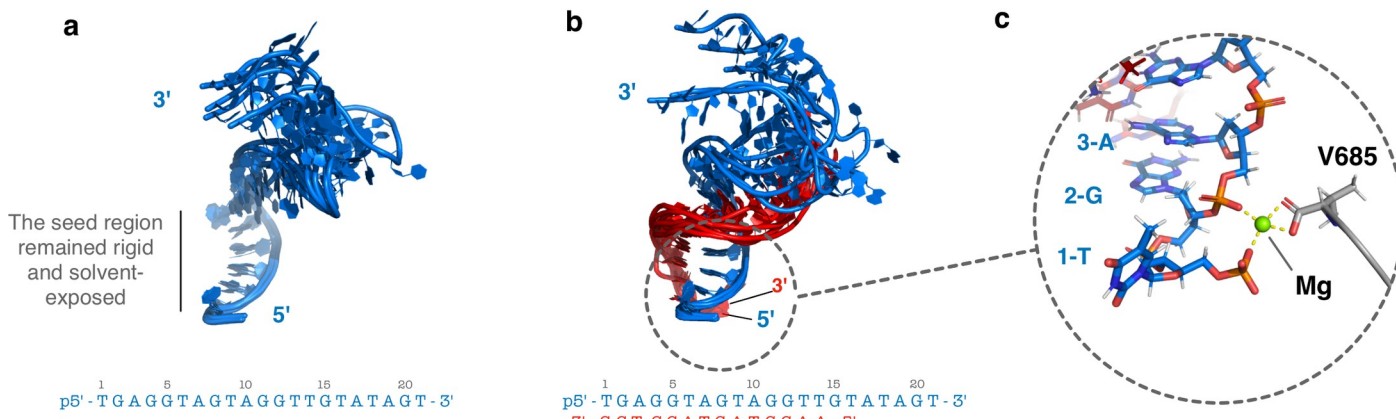

**Fig 4. DNA cluster representatives obtained from the simulation trajectories.** (**a**) Superposition of representative DNA conformations extracted from the 5 most sampled clusters in the simulation of the binary guide-bound TtAgo complex. The seed region remained solvent-exposed in all clusters of the binary complex. (**b**) Same as (a) but for ternary complex (with bound guide strand and target strand). The guide and target strands are displayed in blue and red cartoon respectively. (**c**) The interactions between the Mg ion (green sphere), V685 and the phosphates of the first and the third base remained stable during the simulations.

complex to ~32 Å in the 15-mer and ~46 Å in the 16-mer target complex simulation (**Fig 5**). Secondly, the cluster analysis of the 14-mer target complex indicated that in 64% of the simulation time, the duplex bases at positions 12–14 partially lost base pairing and transiently unstacked from their neighbors–a phenomena termed fraying (**S7 Fig**).

On the contrary, we observed almost no fraying events in the 15-mer and 16-mer target complex (**S8 Fig**). The occurrence of fraying in the 14-mer target complex is further evident from high values of root mean-square atomic fluctuations (RMSF) in the bases 10 to 14 (**Fig 5E**). Overall, there seems to exist an interplay between the N domain and the DNA duplex extension. With 14-mer target, the duplex interacts weakly with the N domain whereas for the 15- and 16-mer targets, the duplex termini gain stability and more extensively contact the N domain, which on average also widens the gap between N and PIWI (**Fig 5A**).

In the inactive TtAgo structures, the 3' end of the guide strand is bound to the PAZ domain. It is known that with the extension of the guide/target duplex beyond position 15, the TtAgo protein adopts an active conformation in which the 3' end is released from the PAZ domain and PAZ has moved towards the MID domain [17]. Interestingly, we observed the dissociation of the 3' end when proceeding to 15- and 16-mer target strands. Cluster analysis of the trajectories showed that in 3% of the frames of the 15-mer target complex the 3' end was released

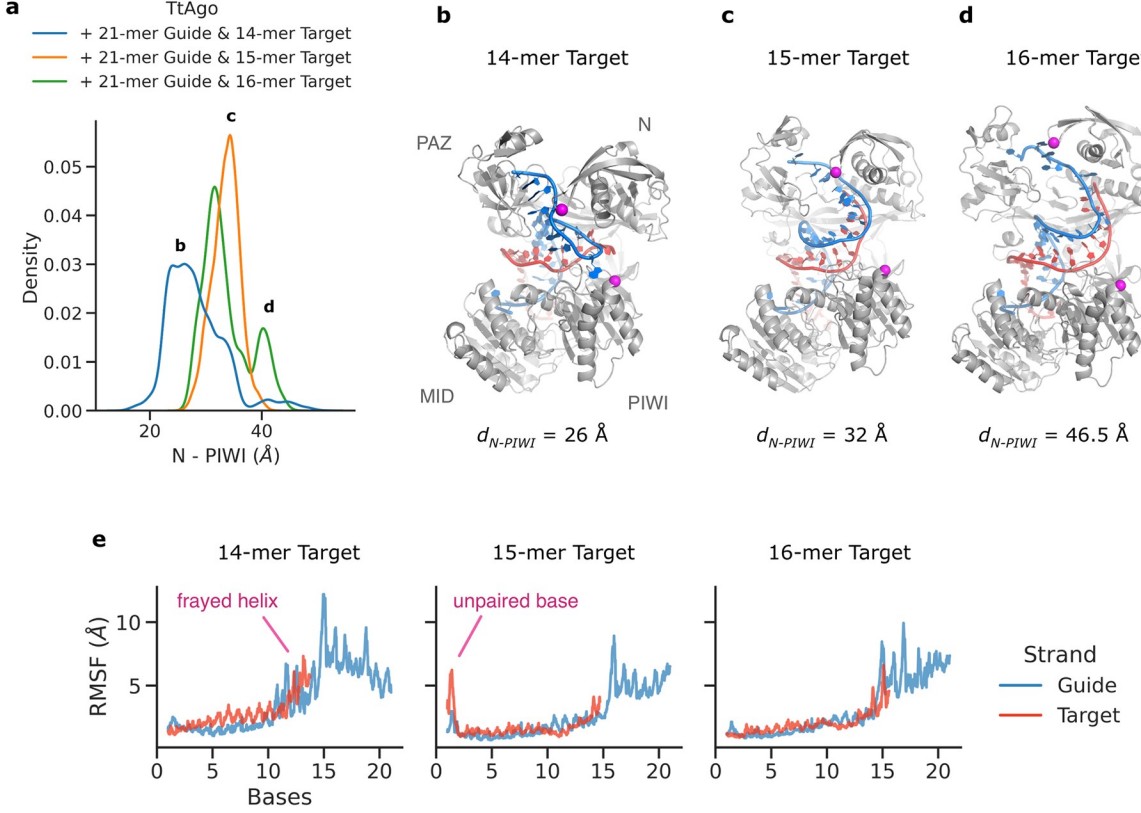

**Fig 5. H-REMD simulations of ternary complexes with 14, 15 or 16 bp between guide and target DNA strands.** (**a**) Sampled distribution of the N-PIWI domain distances in the three ternary complexes Argonaute with varying target length. The distances are measured between α-carbon atoms of the two residues M82 from N and D552 from PIWI domain, indicated as magenta spheres (in panels **b-d**). The distance distributions indicate on average a shift towards N-PIWI opening as the target-guide base pairing is increased to 15 or 16 bp. (**b-d**) This is also evident in snapshots taken from the simulations of 14, 15, and 16-mer target DNA ternary complexes. The guide and target strands are shown in blue and red cartoons). (**e**) Root mean square fluctuations (RMSF) of each base for all the DNA strands during the H-REMD (in the reference replica). The RMSF values were calculated with respect to the average structure.

from PAZ, however still present in the gap between PAZ and N domains (**S8 Fig**). This fraction was increased to 6% in the 16-mer target complex. The release of the 3' end observed in 15-mer and 16-mer complexes explains the increased RMSF values in the guide's 3'-end region in **Fig 5E** compared to the 14-mer target complex. Restarting the 16-mer target simulation from one of these frames resulted in a complete transition of the 3' end towards the N-PIWI gap. Snapshots of the 3' end's release from the PAZ domain are illustrated in **Fig 6A, 6B, 6C and 6D**. The stacking of the aromatic rings of Y43 and P44 over the duplex bases at position 16 was strikingly persistent throughout the release and is likely also responsible for the reduced fraying of the terminal base pairs in case of the bound 15 or 16 bp duplexes. We postulate that this stacking interaction is also a prerequisite for the push back on the N domain, increased distance between N-domain and PIWI domain release of the 3' end from PAZ and subsequent activation of the Argonaute cleavage of the target strand. Our simulation results in accord with the experimental observation that reducing the guide-target duplex's length from 15 to 14 bp strongly diminishes the cleavage of the target[12].

The conformations sampled with the released 3'-end superimpose globally very well with the active TtAgo complex structure (PDB: 4nca), the closest sampled snapshot resulted in a RMSD of < 2 Å (**Fig 6E**). However, the catalytic tetrad in the PIWI domain (as observed in the active TtAgo structure) was not formed during the simulations. This local conformational change may require longer simulations since it is also not part of the CVs and the advanced sampling setup. Next, we measured the distance between a phosphate atom of the 3' end (OP2) and its interacting partner on PAZ (hydroxyl group of Y226). While in the 14-mer target DNA complex, the distance was consistently short (~2.8 Å), it drastically increased in the 15- and 16-mer target variants during the H-REMD simulations (**Fig 6F**). The associated free-energy calculations show that with the extension of the guide/target duplex to position 15 and 16, the energy landscape changes such that the penalty for the 3'-end release decreases. It should be emphasized that the extracted free energy penalty may not be completely converged and may decrease further on longer time scale due to increased accumulation of 3'-end released states.

## Discussion

Experimental high-resolution structures of Argonaute binary and ternary complexes in inactive and active states provide conformational snapshots of the target recognition and catalytic mechanism. These structures indicate movements of Argonaute domains, especially of the PAZ domain, that are associated with guide and target binding as well as transition from inactive to active states. However, putative intermediates between these states and the overall ensemble of possible domain arrangements associated with Argonautes in the apo and various bound structures are not covered by experimental structure determination. In addition, the molecular mechanism of Argonaute activation is not understood in molecular detail.

In our simulation study we propose an enhanced-sampling H-REMD scheme with an adapting biasing potential to systematically study domain motions in Argonaute complexes. Similar to a metadynamics approach the method constructs a biasing potential to increasingly destabilize low free-energy regimes in a series of replicas. In contrast to metadynamics[26] the biasing potential is not added slowly in many steps to the simulation force field, but immediately applied at different levels along the replicas. This leads to rapid exploration of other relevant regimes along the CVs that will be adaptively destabilized and can appear in the unbiased reference replica. An advantage over metadynamics is that instead of using a multi-dimensional biasing potential, each CV is treated by a separate biasing potential. This results in an improved sampling of relevant (low free energy) conformations in the unbiased reference

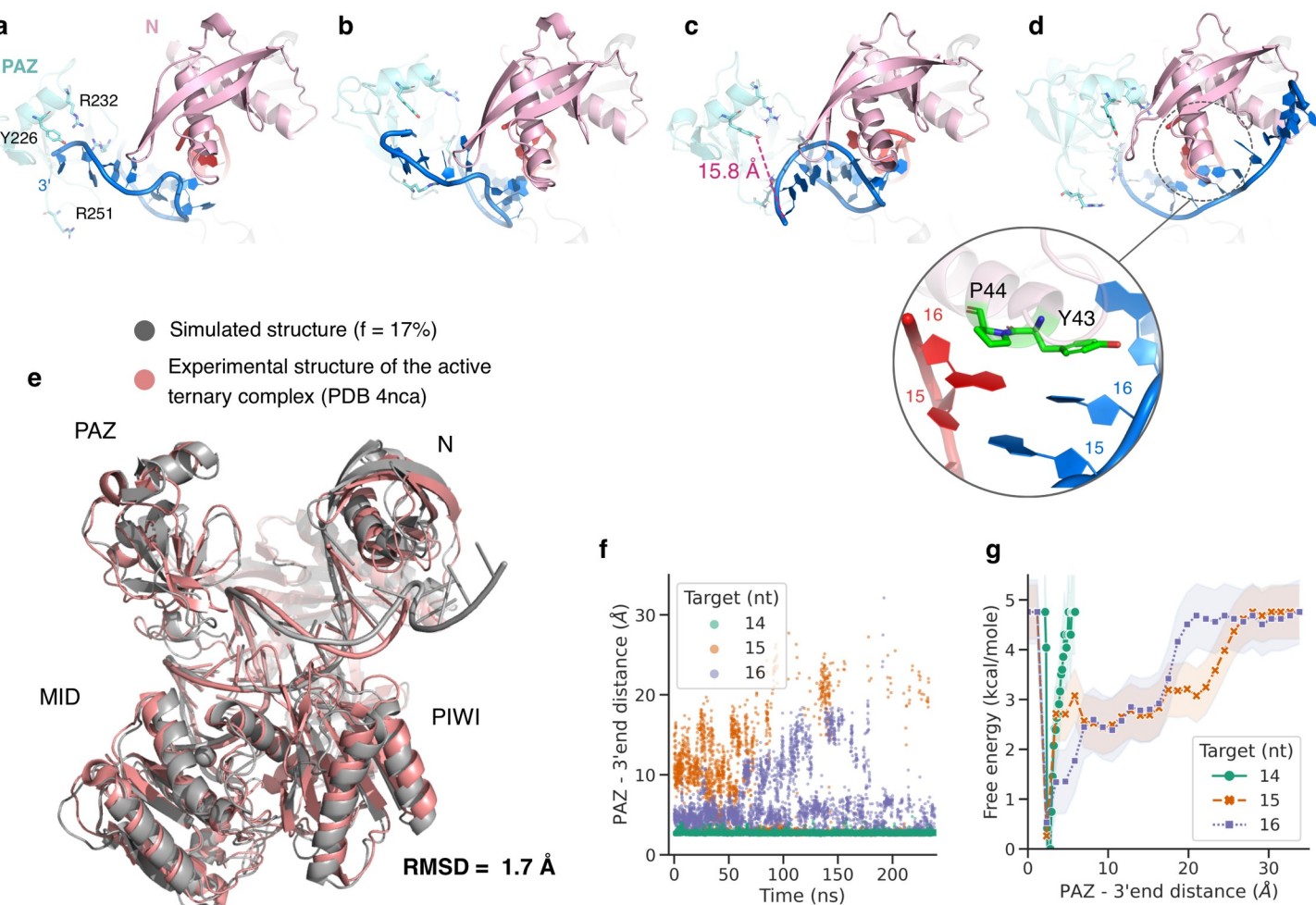

**Fig 6. Guide 3'-end released observed in H-REMD of ternary TtAgo with bound 15 and 16 bp target-guide duplexes.** (**a**-**d**) Simulation snapshots indicating four dynamical states of the release of guide 3' end from the PAZ domain in the ternary complex of TtAgo with 21-mer guide and 16-mer target DNA strands. The guide is represented in blue, the target in red and the PAZ and N domains in cyan and pink respectively. The stacking interactions between Y43 & P44 (green sticks) and guide and target strands at position 16 are persistent throughout the release process. (**e**) Superimposition of a simulated structural snapshot after release of the 3'end of the guide strand shown in grey cartoon on the catalytically active experimental TtAgo structure shown in pink. (**f**) The sampled distance between the hydroxyl group of Y226, located on PAZ, and the guide 3' end (OP2) is plotted for TtAgo in complex with three different target lengths (data from reference replica). (**g**) The sampled distance distribution (Y226 of PAZ vs. guide 3'end allows the estimation of the associated free energy change vs. distance for the three target lengths (error bars are indicated as shaded areas).

replica. Multi-dimensional biasing potentials typically build up very slowly during metadynamics and may require very long simulation times.

The H-REMD scheme was applied to the Argonaute protein, using center-of-mass (COM) distances between domains in a hierarchical manner as CVs. In the early stages of the 2 μs-long unrestrained MD simulations, the apo protein transitioned towards a more compact closed conformation, and afterwards its overall structure fluctuated around this state. In contrast, the H-REMD results showed that the nucleic acid-free apo TtAgo protein can indeed access a broader range of domain arrangements compared to the regular MD results, which includes conformations characterized by an open substrate binding cleft. Such opening of the nucleic acid binding cleft is necessary otherwise the protein cannot readily accommodate the bulky nucleic acid duplexes. In the rubber-band model the energy required to impose the

opening might be compensated by ATP hydrolysis, while the internal tension caused by it drives the subsequent duplex unwinding without consuming ATP [39].

The replica-exchange simulation trajectories also provide considerable insight into the guide- and guide/target-bound complex structures and associated domain mobility. Notably, in the binary complex direct contacts between the PAZ and MID and PIWI domains are formed, which bury the bases 13–17 of the guide DNA in the central cleft. Nevertheless, the seed region of the guide remains exposed to the solvent, which allows probing the target candidates, while adopting a low-energy conformation. In the 14-mer target ternary complex, the cluster analysis results showed that the DNA duplex is frayed in the majority of the times, which is attributed to unfavorable interactions with the N domain. With a frayed duplex, the guide strand lacks the required strain to dissociate from the PAZ pocket. This explains the observations that in the cleavage assays the truncation of the target strand from its 5' end to positions 15 and 14 (relative to the guide strand) sharply reduces the cleavage activity [12].

Duplex propagation in the 15-mer and 16-mer ternary complexes improves its stability, as indicated by the absence of transient fraying events, and more importantly, by the increased tendency of the guide 3'-end to dissociate from PAZ and transition towards adopting the active conformations. In the crystal structure of the 19-mer ternary complex it has been revealed that the N domain blocks the DNA duplex by stacking the aromatic rings of Y43 and P44 on the DNA bases at position 16 [12]. Based on our observations, the stacking interactions seem to form a lever for the duplex to open the N-PIWI channel. In addition, stacking of Y43 on base 16 of the guide strand creates an anchor point for the dissociation of the 3'-end from PAZ and its rotation around the N domain. However, the sampled structures with the released 3'-end appeared to represent cleavage-incompatible conformations as the catalytic tetrad of DEDD amino-acids residues in the PIWI were not transitioned to the active arrangement. Such transition requires additional rearrangements in loops that were not included in our choice of the CVs, but are facilitated by higher temperatures [42]. The H-REMD technique with an adaptive biasing along appropriate CVs to promote domain-domain rearrangements could also be useful to study other multi-domain or multi-subunit proteins and protein-protein complexes that undergo domain motions during functional process cycles.

## Supporting information

**S1 Fig. Root Mean Square Deviation (RMSD) and conformations during cMD simulations of apo Argonoute and binary and ternary complexes.**
(PDF)

**S2 Fig. Definition of the CV for the H-REMD simulations.**
(PDF)

**S3 Fig. Convergence of the H-REMD simulations.**
(PDF)

**S4 Fig. Build up of sampling and biasing potential in H-REMD simulations.**
(PDF)

**S5 Fig. Conformational fluctuations (RMSF).**
(PDF)

**S6 Fig. RMSD vs. simulation time for H-REMD simulations.**
(PDF)

**S7 Fig. Cluster populations and cluster representatives.**
(PDF)

**S8 Fig. Cluster populations and cluster representatives for ternary complexes with 15 and 16 bp guide-targed duplexes.**
(PDF)

## Author Contributions

**Conceptualization:** Martin Zacharias.

**Data curation:** Danial Pourjafar-Dehkordi.

**Formal analysis:** Danial Pourjafar-Dehkordi.

**Funding acquisition:** Martin Zacharias.

**Investigation:** Danial Pourjafar-Dehkordi.

**Methodology:** Martin Zacharias.

**Project administration:** Martin Zacharias.

**Resources:** Martin Zacharias.

**Software:** Martin Zacharias.

**Supervision:** Martin Zacharias.

**Validation:** Danial Pourjafar-Dehkordi.

**Visualization:** Danial Pourjafar-Dehkordi.

**Writing – original draft:** Danial Pourjafar-Dehkordi, Martin Zacharias.

**Writing – review & editing:** Martin Zacharias.

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
