## [Decision Letter · Decision Letter 0]

14 Oct 2021

Dear Dr. Zacharias,

Thank you very much for submitting your manuscript "Binding induced functional domain motions in the argonaute characterized by adaptive advanced sampling" for consideration at PLOS Computational Biology. As with all papers reviewed by the journal, your manuscript was reviewed by members of the editorial board and by several independent reviewers. The reviewers appreciated the attention to an important topic. Based on the reviews, we are likely to accept this manuscript for publication, providing that you modify the manuscript according to the review recommendations.

Based on the insightful comments of the first reviewer, minor revision is recommended. In your revised version, please perform additional simulations so that the statistics for the detachment/reattachment events are significant. Also please address the convergence issue.

Sincerely,

Jeffrey Skolnick

Guest Editor

PLOS Computational Biology

Arne Elofsson

Deputy Editor

PLOS Computational Biology

[LINK]

Based on the insightful comments of the first reviewer, minor revision is recommended. In your revised version, please perform additional simulations so that the statistics for the detachment/reattachment events are significant. Also please address the convergence issue.

Reviewer's Responses to Questions

**Comments to the Authors:**

Reviewer #1: The study of how RNA is processed by cellular machineries is of central importance to nowadays chemical biology. Here Pourjafar-Dehkordi and Zacharias set out to study the recognition between guide RNA and mRNA target regions within the Argonaute protein. The authors carry out a comparative analyis of apo, guide and target bound systems. They use an adaptive sampling scheme based on H-REMD, convincingly showing relevant differences among the sampling in the different states (apo, binary, tertiary).

The paper is well conducted, and the differences in domain motions noticed are well described and based on solid calculations.

I found this paper interesting to read and considering the growing importance of argonaute, it is worth having this paper exposed to the community.

I think the paper is worth publishing as it is: it is very solid work, it contains a large amount of data, and everything is clearly presented.

Reviewer #2: The authors report molecular dynamics simulations of the important Argonaut protein, from which they obtain insights into domain motions, allostery, and other effects. An adaptive sampling method is introduced, which has similaritiies to metadynamics and allows enhanced sampling of slow domain motions. This is valuable as sampling domain motions is a hard and important problem, and the present method is not too complicated. The large and complex protein is simulated alone and with a series of its DNA or RNA substrates. The authors are able to explain its selectivity for nucleic acids above a certain length, which was not understood previously. The simulations and analysis are thorough and mostly well-explained. I have only a few comments on points of clarity.

1) The bias functions are changed every 10 ns; replica exchange then runs for another 10 ns, and so on. Can the authors say something about convergence of the statistical ensemble over this fairly short time segment? In particular, is it correct to say that the unbiased replica obeys its natural distribution of states, domain orientations, and so on, or is it continuously out of equilibrium 80 or 80% of the time?

2) In Fig 6, the distributions of states is converted into a free energy distribution, which describes the detachment of the nucleotide 3' terminus from the protein. Although the main result is clear (stability of the 14-mer), the free energis for the 15- and 16-mer rely on just a few detachment/reattachment events: around 10 for the 16-mer and just a handful for the 15-mer. Also, the 15-mer data are unclear, and seem to be hidden by the 14-mer points. Can the authors provide an error bar for the free energy curves and clarifiy the figure a bit?

3) There is a "Results and Discussions" (sic) section and also a "Discussion" section. Probably this should be fixed.

Overall, there is a good level of novelty, a large and careful data set, and a number of interesting insights into the protein function. Experimental validation is mainly limited to the detachment of the longer nucleotide strands.

Reviewer #3: The manuscript is an tour-de force description of a new approach to study Argonaute domain motion using simulations. It, unfortunately, is not very accessible to the non-expert. This is unfortunate, because the subject itself might be of interest to a wider audience (than biophysicists), too. The manuscript mentions "miRNAs" twice in the abstract and nowhere else after, which is strange and should be addressed by making clear how miRNAs are different from siRNAs and how AGO in non-miRNA organisms, such as Bacteria, act totally different from AGOs that load miRNAs and their targets (a discussion of target directed miRNA decay could be interesting).

**Have the authors made all data and (if applicable) computational code underlying the findings in their manuscript fully available?**

Reviewer #1: Yes

Reviewer #2: Yes

Reviewer #3: None

PLOS authors have the option to publish the peer review history of their article (what does this mean?). If published, this will include your full peer review and any attached files.

Reviewer #1: No

Reviewer #2: No

Reviewer #3: No

Figure Files:

Data Requirements:

Reproducibility:

References:

---

## [Editor Report · Decision Letter 1]

9 Nov 2021

Dear Dr. Zacharias,

We are pleased to inform you that your manuscript 'Binding induced functional domain motions in the argonaute characterized by adaptive advanced sampling' has been provisionally accepted for publication in PLOS Computational Biology.

Best regards,

Jeffrey Skolnick

Guest Editor

PLOS Computational Biology

Arne Elofsson

Deputy Editor

PLOS Computational Biology

The revisions address all the minor concerns of the reviewers and myself. The paper is now acceptable for publication. Congratulations on a very nice piece of work.

---

## [Editor Report · Acceptance letter]

22 Nov 2021

PCOMPBIOL-D-21-01683R1 

Binding induced functional domain motions in the argonaute characterized by adaptive advanced sampling

Dear Dr Zacharias,

I am pleased to inform you that your manuscript has been formally accepted for publication in PLOS Computational Biology. Your manuscript is now with our production department and you will be notified of the publication date in due course.

With kind regards,

Agnes Pap
